# Performance Zoning of Asphalt Pavement and Performance Grade (PG) of Asphalt Binder in Karamay: A Case Study of Xinjiang, China

Chaofei Dong [1], Liqun Feng [1,2,3,*] and Yafeng Xu [2,3]

1.  College of Architecture and Engineering, Xinjiang University, Urumchi 830017, China
2.  Xinjiang Academy of Transportation Research, Urumchi 830099, China
3.  Key Laboratory of Highway Engineering Technology in Arid Desert Zone Transportation Industry, Urumchi 830000, China
*   Correspondence: 15735103919@163.com; Tel.: +86-15735103919

**Abstract:** Asphalt binder is a temperature-sensitive material with a performance that is greatly affected by changing climates. Improper selection of asphalt will cause a lot of damage and affect the durability of the road. The establishment of asphalt pavement performance zoning in Xinjiang, a vast area with great temperature differences, will provide a reference for the selection of asphalt suitability, the refinement of pavement design, and the sustainable development of road engineering. In this study, 11 years of temperature data in the Xinjiang region have been collected and analyzed, and 98% reliability of pavement design temperature has been used to draw a performance grading map of asphalt pavement in the Xinjiang region based on the ArcGIS platform. Finally, the Xinjiang region is divided into nine performance zones. At the same time, the performance grades (PG) of five kinds of asphalt binders in Karamay are explored. The result shows that there is little difference in continuous PG span between different matrix asphalt binders; the lower the penetration grade, the better the high-temperature performance, and the worse the low-temperature performance. After adding the SBS modifier, the continuous PG span can be about 20 °C higher than the matrix asphalt. The indoor test of asphalt mixture also shows that SBS-modified asphalt has better durability. All these provide the basis for a reasonable selection of asphalt binders in different areas of Xinjiang.

**Keywords:** performance zoning; Superpave; performance grade; dynamic shear rheometer; bending beam rheometer

## 1. Introduction

Asphalt pavement has the advantages of low noise, easy maintenance, and fast construction and is widely used in most high-grade roads in China [1–3]. Xinjiang is a vast territory, and the climate varies greatly from region to region, which greatly tests the durability of asphalt pavement in the Xinjiang region. Various factors influence asphalt pavement durability on roads because the construction processes, quality of materials, design aspects, weather conditions, as well as operating conditions directly contribute to asphalt pavement durability. Most of the studies where factors that impact asphalt pavements durability are identified mainly focus on (1) traffic effects and vehicular loads, (2) characteristics and properties of the materials that make up the structure of the pavements, and (3) weather and road operation effects [4]. Under the coupled action of traffic and environmental effects, asphalt binders undergo significant physical changes such as aging, healing, and premature failure due to their sensitivity to temperature and loading rate, resulting in the early deterioration of asphalt pavement performance [5–7]. The climatic factors that affect the performance of asphalt pavements include high temperature, low temperature, water, fatigue, and aging [8,9]. Asphalt pavements are prone to rutting lesions

under high-temperature driving loads [10], fatigue cracks under intermediate temperatures [11], experiencing prolonged repetitive driving loads, and transverse cracks under low-temperature cold environments [12,13], all of which seriously affect driving safety. The rise of pavement temperature due to global climate change over recent years creates higher requirements for the high- and low-temperature performance of asphalt roads [14].

The performance of asphalt pavement mainly depends on the performance of the asphalt binder material used for construction [15]. Asphalt binder has both viscous and elastic properties, showing soft viscosity at high temperatures and brittle behavior at low temperatures [16]. Increasing the modulus of the asphalt binder can improve the load-bearing capacity of the pavement, thus reducing the occurrence of rutting [17–20]. There are usually two ways to increase the modulus of asphalt binder: one is to use asphalt with a lower penetration grade, such as 50# or 30# matrix asphalt binder, and the second is to add modified substances in the asphalt binder, such as SBS modifier and various high modulus agents [21]. Styrene-butadiene-styrene (SBS), as a copolymer, is often used as a modifier for asphalt binders to improve the bonding properties of the binder [22]. SBS modifiers can form a cross-linked network that improves the flexibility, elasticity, and durability of the binder; this allows the asphalt to better resist cracking, rutting, and other forms of damage, thereby extending the life of the pavement [23]. In addition, the SBS modifier improves the viscosity of the asphalt, which results in better binding to the aggregate to form a more durable pavement [24]. However, the use of modified asphalt will greatly increase production costs, intensify the consumption of non-renewable resources, aggravate air pollution, and is not conducive to the sustainable development of the local economy [25,26]. Xinjiang has abundant resources of low penetration asphalt grade, and the production cost is low. Due to the lack of comprehensive understanding of the performance of asphalt with low penetration grade, the 90# matrix asphalt is still used for most of the asphalt pavement in Xinjiang. In summary, it is necessary to research the performance law of matrix asphalt with different penetration grades and the gap between them and SBS-modified asphalt.

At present, the standard asphalt classification in China is mainly based on the penetration grade; this method is convenient but cannot accurately assess the high- and low-temperature performance of asphalt [27]. In contrast, the characterization methods based on performance grading can better simulate the actual conditions to which the asphalt is subjected [28,29].

The asphalt binder performance grade (PG) system based on asphalt rheological properties is a product of the Strategic Highway Research Program (SHRP) in the United States [30]. In Superpave research, rheological analysis is widely used for road paving asphalt, including Dynamic Shear Rheometer (DSR) and Bending Beam Rheometer (BBR) [31]. Among them, the DSR test can provide a large amount of high-temperature performance information, such as composite shear modulus (G*), phase angle ($\delta$), rutting factor (G*/sin$\delta$), and fatigue factor (G*·sin$\delta$); these parameters are used to quantify the high-temperature rutting resistance and medium-temperature fatigue resistance of asphalt binder [32]. The creep stiffness modulus S and m values obtained from the BBR test are characterized by the low-temperature performance of asphalt.

According to the actual environment in which the asphalt pavement is located, the selection of suitable asphalt binder can not only enhance the durability [33,34], comfort, and safety of asphalt pavement but also achieve the purpose of economic and environmental protection, which is an effective way to achieve sustainable development. Therefore, many countries have conducted climate-based performance zoning studies and developed performance-level zoning standards for asphalt pavements based on local climate as a reference for asphalt suitability selection so that asphalt pavement design can be adapted to the climatic environment of a region [35,36]. For example, the U.S. Strategic Highway Research Program (SHRP) proposed a method for classifying asphalt pavement performance grades (PG) based on high- and low-temperature indicators [37]. Asi et al. proposed a temperature–climate zoning method that divided Jordan into three zones based on the high and low temperatures of the pavement at a 98% confidence level [38]. Hassam et al. used a

data development model to predict high and low temperatures of asphalt pavements in Oman and proposed performance grades of asphalt binders for each region in Oman [39]. Saleh et al. converted road surface and pavement temperatures based on Superpave and LTTP projects to generate four asphalt performance grading zones for Egypt [40]. Mirza et al. recommend an SHRP model with 98% reliability that divides Pakistan into six PG grading zones [41]. Jitsangiam et al. divided northern Thailand into two grading zones by calculating the average temperature and standard deviation value at a 95% confidence level [42]. Salem et al. used the SHRP model with 50% reliability to classify the road PG grading of the Libyan desert into three zones [43]. Viola et al. developed an isoline map of pavement temperature in Italy based on the Superpave specification and considered the effects of climate change on pavement performance [44]. Cota et al. generated a PG grading chart of asphalt binders in Mexico based on temperature and elevation, which was used to determine the grade of asphalt binders required [44]. Zhang et al. proposed a temperature conversion formula for asphalt pavements in northeast China based on the SHRP method, established a PG climate zone in northeast China, and evaluated the high- and low-temperature performance of asphalt binders [15]. Zhao et al. proposed performance zoning indexes for different asphalt pavement layers in Inner Mongolia, China, and divided the asphalt pavement in Inner Mongolia into three main performance zones and six secondary performance zones and verified them [45]. To sum up, the selection of asphalt is inseparable from the guidance of the performance zone of asphalt pavement, and it is necessary to establish the pavement performance grading map in the Xinjiang region.

This paper is mainly focused on the collection and analysis of meteorological data in different regions of Xinjiang to determine the environment in which the asphalt is located by pavement temperature, to establish a performance grade zoning chart of asphalt pavement in Xinjiang, and to propose the high- and low-temperature performance of asphalt to be used in different regions. Due to the wide application of Karamay asphalt in the Xinjiang region, four different penetration grades of matrix asphalt and one SBS-modified asphalt are selected for the PG grading study. The difference between high- and low-temperature properties of matrix asphalt with different penetration grades and softening point temperatures are analyzed; this also includes the gap between matrix asphalt and modified asphalt. These findings will provide a reference and basis for the suitable selection of asphalt binders in the Xinjiang region and also points out the direction for the sustainable development of asphalt pavement in the Xinjiang region.

Based on the research background and research objectives, a flowchart is given in Figure 1 to show the research content and conclusion of this paper.

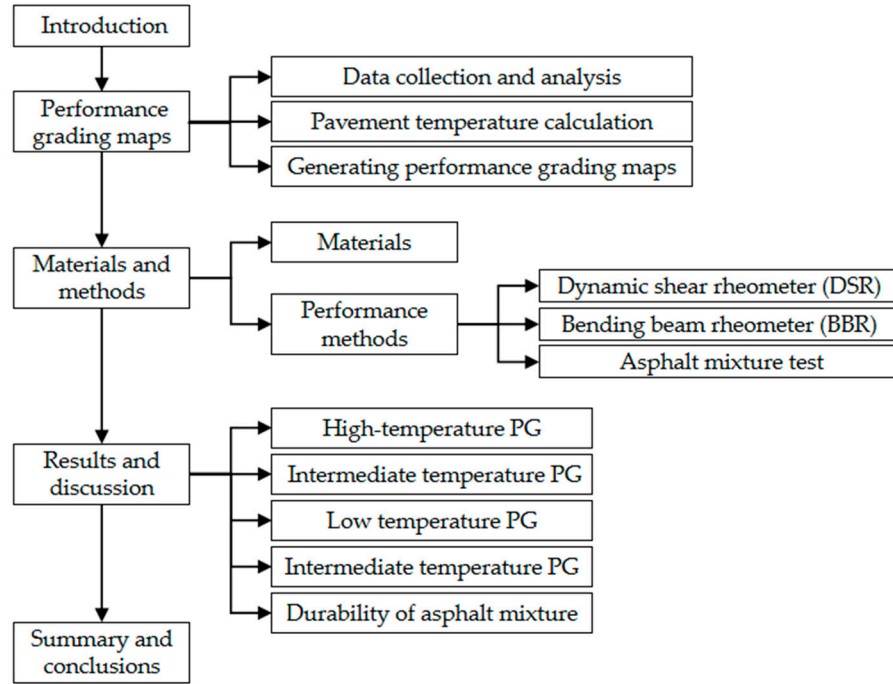

**Figure 1.** The flowchart of this paper.

**2. Generation Performance Grading Map of Asphalt Pavement for Xinjiang**

*2.1. Temperature Data Collection and Analysis*

The performance grade of Superpave asphalt binder is established based on the minimum and maximum pavement temperature expected on site. Due to the lack of pavement temperature data in Xinjiang, data were obtained by proposing a conversion model based on the relationship between air temperature and pavement temperature. Therefore, the latitude and longitude of 84 stations in Xinjiang were collected and analyzed, as well as the average values of summer maximum temperature for seven consecutive days and extremely low temperature for seven consecutive days throughout the year for the past 11 years.

If the calculations are performed as an average of the temperatures, only 50% reliability levels of high and low pavement temperatures can be obtained. Reliability is the percentage probability that the actual temperature does not exceed the design temperature over the course of a year; a higher percentage means lower risk. The maximum and minimum design temperatures at 98% reliability are calculated statistically. A reliability of 98% is two standard deviations from the average value. The calculations are as follows:

$$T(max) \text{ at } 98\% = X(\text{High Temp}) + 2 \times S(\text{High Temp})$$

$$T(min) \text{ at } 98\% = X(\text{Low Temp}) - 2 \times S(\text{Low Temp})$$

where the standard deviation value is determined by the calculation of Equation (1).

$$S = \sqrt{\frac{1}{N-1}\sum_{i=1}^{n}(X_i - X)} \tag{1}$$

where $X_i$ is a single temperature record, $X$ is the average value of a set of temperatures, and $N$ is the number of temperature records in a set.

These values are listed in Table 1, and the standard deviation values of maximum temperatures in different regions ranged from 1.1–2.8, while the standard deviation values of minimum temperatures ranged from 1.8–4.7, indicating that the maximum temperatures

in the hot season had little difference in Xinjiang over the past decade, while the minimum temperatures fluctuated more in the cold season.

**Table 1.** Maximum and minimum temperatures and pavement design temperatures for stations in Xinjiang.

| Station | Longitude (°) | Latitude (°) | Average of Seven Days, Average Maximum Temperature (°C) | Standard Deviation | Maximum Design Temperature of Road Surface (°C) | Average of Seven Days, Average Minimum Temperature (°C) | Standard Deviation | Minimum Design Temperature of Road Surface (°C) |
|---|---|---|---|---|---|---|---|---|
| Fuhai | 87.48 | 47.11 | 38.0 | 2.2 | 65.50 | −39.0 | 3.1 | −31.83 |
| Habahe | 86.32 | 48.04 | 37.4 | 2.4 | 64.99 | −37.8 | 4.1 | −30.80 |
| Buerjin | 86.85 | 47.70 | 37.4 | 2.4 | 64.97 | −39.0 | 3.8 | −31.83 |
| Jimunai | 85.87 | 47.44 | 35.4 | 2.8 | 63.05 | −34.8 | 4.1 | −28.22 |
| Fuyun | 89.53 | 46.99 | 38.4 | 2.8 | 65.88 | −41.0 | 4.1 | −33.55 |
| Qinghe | 90.38 | 46.67 | 35.0 | 2.2 | 62.61 | −42.2 | 3.4 | −34.58 |
| Tacheng | 82.59 | 46.46 | 38.2 | 1.9 | 65.65 | −33.2 | 4.1 | −26.84 |
| Emin | 83.62 | 46.52 | 36.5 | 2.0 | 64.03 | −36.5 | 4.5 | −29.68 |
| Yumin | 82.98 | 46.20 | 38.0 | 2.2 | 65.44 | −34.2 | 4.4 | −27.70 |
| Tuoli | 83.60 | 45.94 | 34.8 | 1.9 | 62.37 | −31.4 | 3.6 | −25.29 |
| Hebukesaier | 85.72 | 46.79 | 32.2 | 1.5 | 59.95 | −34.8 | 3.8 | −28.22 |
| Wusu | 84.62 | 44.45 | 38.9 | 1.7 | 66.17 | −30.6 | 2.5 | −24.61 |
| Kelamayi | 84.77 | 45.59 | 39.5 | 1.7 | 66.83 | −32.3 | 3.2 | −26.07 |
| Kuitun | 84.89 | 44.45 | 38.9 | 1.6 | 66.17 | −30.4 | 2.5 | −24.43 |
| Shawan | 85.62 | 44.33 | 37.7 | 1.3 | 65.02 | −31.5 | 3.1 | −25.38 |
| Manasi | 86.20 | 44.29 | 39.1 | 1.8 | 66.35 | −33.1 | 2.8 | −26.76 |
| Shihezi | 86.00 | 44.18 | 39.2 | 1.5 | 66.44 | −38.5 | 4.6 | −31.40 |
| Changji | 87.30 | 44.02 | 40.3 | 2.1 | 67.48 | −37.1 | 3.8 | −30.19 |
| Wulumuqi | 87.61 | 43.79 | 37.7 | 2.1 | 64.98 | −29.4 | 3.0 | −23.58 |
| Wujiaqu | 87.54 | 44.17 | 41.0 | 2.1 | 68.16 | −36.8 | 2.6 | −29.94 |
| Fukang | 87.94 | 44.16 | 40.2 | 1.8 | 67.39 | −32.5 | 3.0 | −26.24 |
| Miquan | 87.68 | 43.97 | 40.0 | 1.9 | 67.19 | −33.0 | 4.0 | −26.67 |
| Qitai | 89.59 | 44.02 | 38.2 | 1.7 | 65.47 | −37.1 | 3.3 | −30.19 |
| Jimusaer | 89.18 | 44.00 | 38.2 | 1.6 | 65.47 | −32.2 | 3.2 | −25.98 |
| Mulei | 90.28 | 43.83 | 34.4 | 1.8 | 61.83 | −32.6 | 3.9 | −26.33 |
| Balikun | 93.01 | 43.59 | 33.0 | 2.0 | 60.48 | −35.0 | 3.3 | −28.39 |
| Yiwu | 94.69 | 43.25 | 32.8 | 1.7 | 60.26 | −31.3 | 2.6 | −25.21 |
| Hami | 93.44 | 42.78 | 41.4 | 1.4 | 68.43 | −26.6 | 2.6 | −21.17 |
| Tulufan | 89.18 | 42.93 | 46.2 | 1.3 | 73.02 | −19.8 | 2.6 | −15.32 |
| Shanshan | 90.21 | 42.86 | 43.8 | 1.4 | 70.73 | −22.5 | 2.2 | −17.64 |
| Tuokesun | 88.65 | 42.79 | 46.5 | 1.6 | 73.30 | −20.4 | 2.0 | −15.84 |
| Hejing | 86.39 | 42.31 | 36.7 | 1.3 | 63.90 | −24.2 | 1.9 | −19.10 |
| Heshuo | 86.86 | 42.26 | 36.6 | 1.2 | 63.80 | −25.6 | 2.2 | −20.31 |
| Yanqi | 86.57 | 42.05 | 37.2 | 1.5 | 64.36 | −26.0 | 2.6 | −20.65 |
| Bohu | 86.63 | 41.98 | 36.5 | 1.1 | 63.69 | −25.6 | 2.7 | −20.31 |
| Kuerle | 86.06 | 41.68 | 38.7 | 1.7 | 65.76 | −20.8 | 1.9 | −16.18 |
| Yuli | 86.25 | 41.33 | 39.3 | 1.3 | 66.30 | −24.3 | 2.5 | −19.19 |
| Ruoqiang | 88.17 | 39.02 | 42.3 | 1.5 | 68.96 | −22.3 | 1.9 | −17.47 |
| Qiemo | 85.53 | 38.14 | 40.2 | 1.4 | 66.87 | −21.6 | 1.8 | −16.87 |
| Minfeg | 82.68 | 37.06 | 40.6 | 1.7 | 67.15 | −21.1 | 2.4 | −16.44 |
| Yutian | 81.95 | 36.45 | 39.2 | 1.4 | 65.75 | −19.7 | 2.3 | −15.24 |
| Cele | 80.78 | 37.04 | 40.0 | 1.5 | 66.57 | −19.2 | 2.3 | −14.81 |
| Hetian | 79.94 | 37.12 | 39.1 | 1.5 | 65.72 | −18.4 | 2.7 | −14.12 |
| Moyu | 79.71 | 37.31 | 38.9 | 1.3 | 65.55 | −20.8 | 2.1 | −16.18 |
| Pishan | 78.29 | 37.62 | 39.3 | 1.5 | 65.96 | −19.9 | 2.5 | −15.41 |
| Tashiuergan | 75.23 | 37.77 | 30.2 | 1.5 | 57.29 | −28.7 | 1.8 | −22.97 |
| Yecheng | 77.42 | 37.89 | 37.7 | 1.4 | 64.46 | −20.0 | 2.8 | −15.49 |
| Zepu | 77.26 | 38.20 | 38.5 | 1.4 | 65.25 | −20.1 | 2.3 | −15.58 |
| Shache | 77.25 | 38.45 | 38.8 | 1.5 | 65.56 | −19.5 | 2.5 | −15.06 |
| Maigaiti | 77.64 | 38.95 | 39.1 | 1.4 | 65.90 | −19.8 | 2.3 | −15.32 |
| Yingjisha | 76.17 | 38.93 | 38.6 | 1.2 | 65.42 | −20.8 | 3.1 | −16.18 |
| Yuepuhu | 76.77 | 39.23 | 40.1 | 1.3 | 66.88 | −20.7 | 2.5 | −16.10 |
| Jiashi | 76.73 | 39.50 | 39.4 | 1.3 | 66.24 | −20.6 | 2.5 | −16.01 |
| Shule | 76.05 | 39.41 | 38.3 | 1.5 | 65.18 | −19.5 | 2.5 | −15.06 |

**Table 1.** *Cont.*

| Station | Longitude (°) | Latitude (°) | Average of Seven Days, Average Maximum Temperature (°C) | Standard Deviation | Maximum Design Temperature of Road Surface (°C) | Average of Seven Days, Average Minimum Temperature (°C) | Standard Deviation | Minimum Design Temperature of Road Surface (°C) |
|---|---|---|---|---|---|---|---|---|
| Shufu | 75.86 | 39.37 | 37.9 | 1.2 | 64.79 | −19.5 | 2.6 | −15.06 |
| Aketao | 75.95 | 39.15 | 37.6 | 1.2 | 64.48 | −21.8 | 2.8 | −17.04 |
| Kashi | 75.99 | 39.46 | 37.8 | 1.3 | 64.70 | −20.6 | 2.8 | −16.01 |
| Wuqia | 75.25 | 39.71 | 32.4 | 1.5 | 59.57 | −23.6 | 1.9 | −18.59 |
| Atushi | 76.16 | 39.73 | 39.0 | 1.4 | 65.87 | −18.5 | 2.5 | −14.20 |
| Bachu | 78.59 | 39.78 | 39.0 | 1.2 | 65.88 | −20.0 | 2.2 | −15.49 |
| Tumukeshu | 79.13 | 39.85 | 39.2 | 1.3 | 66.08 | −16.4 | 2.7 | −12.40 |
| Aheqi | 78.44 | 40.93 | 33.9 | 1.6 | 61.11 | −24.2 | 2.2 | −19.10 |
| Keping | 79.05 | 40.51 | 38.7 | 1.2 | 65.66 | −22.8 | 2.8 | −17.90 |
| Awati | 80.37 | 40.64 | 37.8 | 1.6 | 64.81 | −21.5 | 2.3 | −16.78 |
| Wushi | 79.22 | 41.21 | 35.1 | 1.2 | 62.28 | −24.4 | 2.9 | −19.28 |
| Wensu | 80.24 | 41.27 | 37.6 | 1.4 | 64.67 | −22.0 | 2.6 | −17.21 |
| Akesu | 80.26 | 41.17 | 37.9 | 1.2 | 64.95 | −21.5 | 2.2 | −16.78 |
| Alaer | 81.29 | 40.54 | 38.4 | 1.2 | 65.37 | −23.5 | 2.7 | −18.50 |
| Shaya | 83.19 | 41.05 | 39.0 | 1.5 | 65.99 | −20.8 | 2.5 | −16.18 |
| Kuche | 82.96 | 41.71 | 37.4 | 1.3 | 64.52 | −21.9 | 2.5 | −17.13 |
| Xinhe | 82.63 | 41.55 | 37.6 | 1.3 | 64.70 | −22.6 | 2.5 | −17.73 |
| Baicheng | 81.84 | 41.82 | 36.5 | 1.4 | 63.67 | −27.0 | 3.2 | −21.51 |
| Luntai | 84.25 | 41.77 | 39.1 | 1.5 | 66.15 | −24.3 | 3.0 | −19.19 |
| Zhaosu | 81.13 | 43.15 | 29.8 | 1.8 | 57.39 | −30.1 | 3.0 | −24.18 |
| Tekesi | 81.83 | 43.21 | 34.3 | 1.5 | 61.69 | −29.6 | 3.3 | −23.75 |
| Gongliu | 82.23 | 43.48 | 36.7 | 1.5 | 64.00 | −30.1 | 3.5 | −24.18 |
| Xinyuan | 83.26 | 43.42 | 36.5 | 1.5 | 63.80 | −25.3 | 3.4 | −20.05 |
| Nileke | 82.51 | 43.80 | 35.9 | 1.6 | 63.26 | −32.0 | 3.5 | −25.81 |
| Yining | 81.33 | 43.91 | 38.2 | 1.7 | 65.46 | −31.2 | 4.2 | −25.12 |
| Chabuchaer | 81.15 | 43.84 | 38.7 | 1.6 | 65.94 | −33.7 | 4.7 | −27.27 |
| Huocheng | 80.87 | 44.05 | 38.7 | 1.6 | 65.95 | −30.7 | 3.7 | −24.69 |
| Wenquan | 81.03 | 44.97 | 33.4 | 1.9 | 60.96 | −31.6 | 2.0 | −25.47 |
| Bole | 82.10 | 44.93 | 38.8 | 1.5 | 66.11 | −32.8 | 2.7 | −26.50 |
| Jinghe | 82.88 | 44.60 | 39.5 | 1.5 | 66.76 | −33.2 | 2.7 | −26.84 |

Since the temperature data obtained with 98% reliability cannot cover all areas, the spatial difference method is generally used to predict the characteristics of unknown geographical areas by using known partial spatial information, which is also the biggest advantage of the spatial difference method. The inverse distance weighting (IDW) method, as a fast and accurate deterministic interpolation method, can calculate weights from the distances of known and unknown points. In space, things that are closer to each other are more similar than things that are farther away from each other. When predicting values for any unmeasured location, the inverse distance weighting method uses the measurements around the predicted location. The measurements closest to the predicted location are assumed to have a greater influence on the predicted value than measurements farther away from the predicted location. The function is

$$Z = \frac{\sum_{i=1}^{n} \frac{1}{D_i^p} Z_i}{\sum_{i=1}^{n} \frac{1}{D_i^p}} \tag{2}$$

where $Z$ is the interpolation point value, $Z_i$ is the observation value of the $i$th sample point, $D_i^p$ is the distance between the ith observation point and the interpolation point, $n$ is the number of samples, and $p$ is the power of the distance.

In ArcGIS, the longitude and latitude of 84 meteorological points and 98% reliability of temperature data were input, and the IDW spatial interpolation method was used to obtain the high-temperature isotherm and low-temperature isotherm in Xinjiang, respectively, as shown in Figure 2a,b. The higher temperatures in different areas of Xinjiang under 98% reliability range from 29.8 °C to 46.5 °C, and the lower temperatures range from −16.4 °C

to −42.2 °C. Most of the areas are under the climate conditions of hot summer and cold winter.

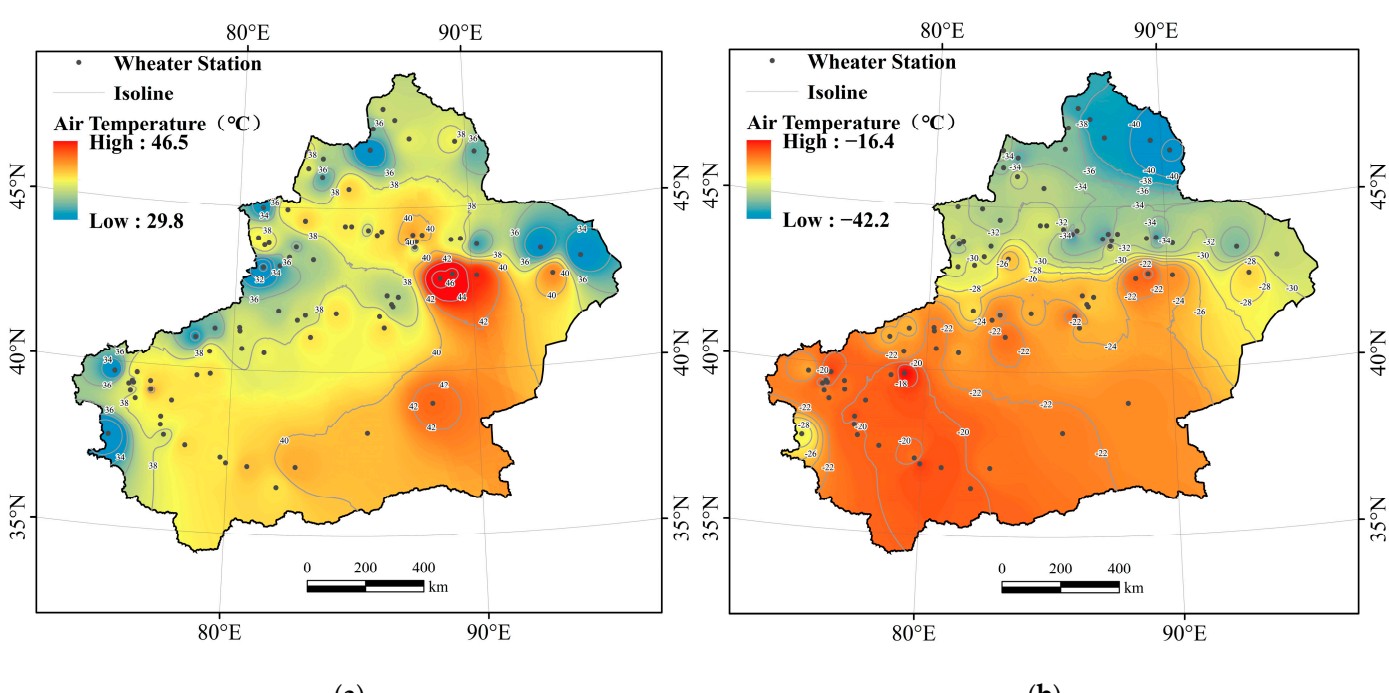

(**a**)                    (**b**)

**Figure 2.** Temperature map with 98% reliability in the Xinjiang region. (**a**) High-temperature isotherm map; (**b**) low-temperature isotherm map.

### 2.2. Pavement Temperature Calculation

According to the theory of net heat flow and energy balance, the road temperature conversion model in Xinjiang is proposed in the Xinjiang Highway Asphalt Pavement Design Instruction Manual. The maximum air temperature is converted to the maximum design temperature at 20 mm below the road surface as defined by Formula (2), and the minimum air temperature is converted to the minimum design temperature of the road surface as defined by Formula (3).

$$T_{20mm} = \left( T_{air} - 0.00618 Lat^2 + 0.2289 Lat + 42.2 \right) \times 0.9545 - 17.78 \tag{3}$$

where $T_{20mm}$ is the high-temperature design temperature of the pavement at 20 mm below the road surface, °C; $T_{air}$ is the maximum temperature with 98% reliability, °C; and *Lat* is the geographic dimension, °.

$$T_{min} = 0.8597 T_{air} + 1.7 \tag{4}$$

where $T_{min}$ is the low-temperature design temperature of the road surface, °C, and $T_{air}$ is the minimum temperature with 98% reliability, °C.

By inputting the high- and low-temperature data with 98% reliability into the pavement temperature conversion model, the maximum and minimum design temperatures of the pavement can be obtained. The specific temperature values are listed in Table 1. The isotherms of maximum temperature and minimum temperature of pavement temperature were generated, respectively, as shown in Figure 3a,b. The extreme maximum temperature reached 73.3 °C in some areas, the minimum extreme maximum temperature was 57.3 °C, and the extreme minimum temperature of pavement was 34.6 °C.

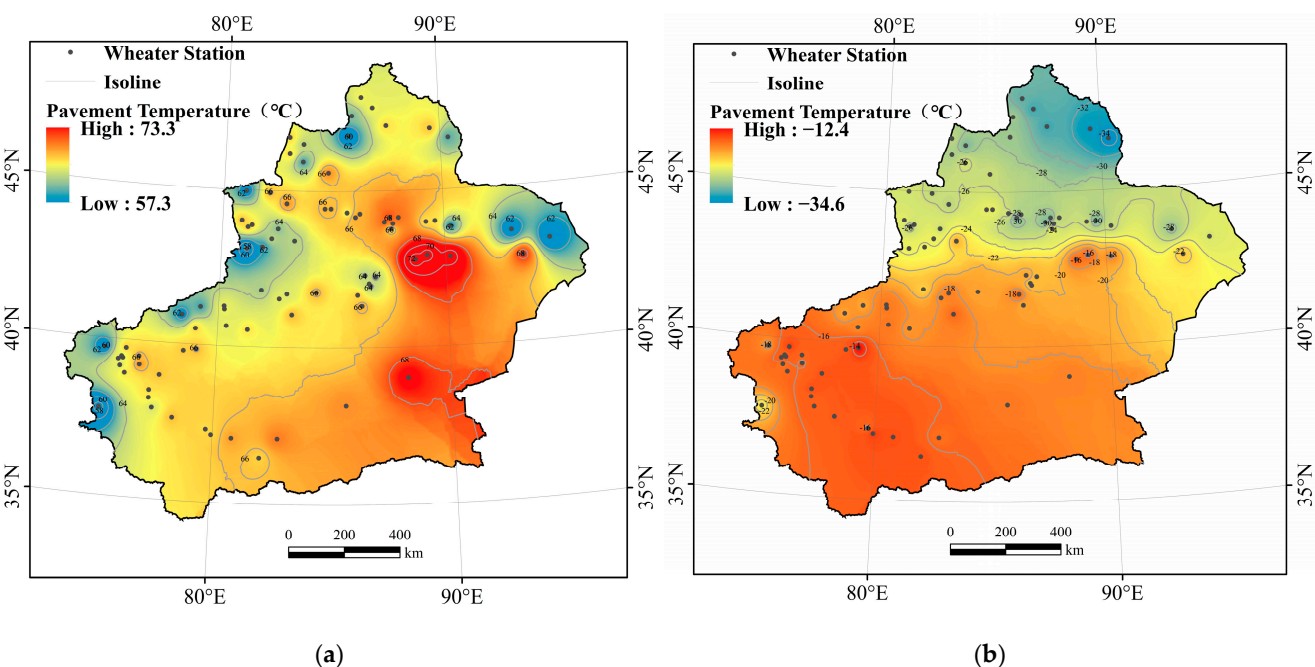

(**a**)                                                                                          (**b**)

**Figure 3.** Pavement temperature map with 98% reliability in the Xinjiang region. (**a**) High-temperature isotherm map; (**b**) low-temperature isotherm map.

### 2.3. Generating Performance Grading Map for Xinjiang

The performance grade of asphalt pavement in Xinjiang was drawn by analyzing the pavement temperature, and finally, nine zones of PG grading were divided, which are PG76-22, PG70-16, PG70-22, PG70-28, PG70-34, PG64-22, PG64-28, PG64-34, and PG58-40, as shown in Figure 4. As can be seen from Figure 4, the four divisions with the largest area share are PG70-16, PG70-22, PG70-28, and PG70-34, indicating that the pavement temperature can reach 70 °C in most areas of Xinjiang during the hot season. The asphalt used in different performance zones in Figure 4 needs to meet the corresponding requirements of maximum and minimum design temperatures so as to ensure the durability and reliability of the road; it also provides a basis for the refined design of asphalt pavements in Xinjiang.

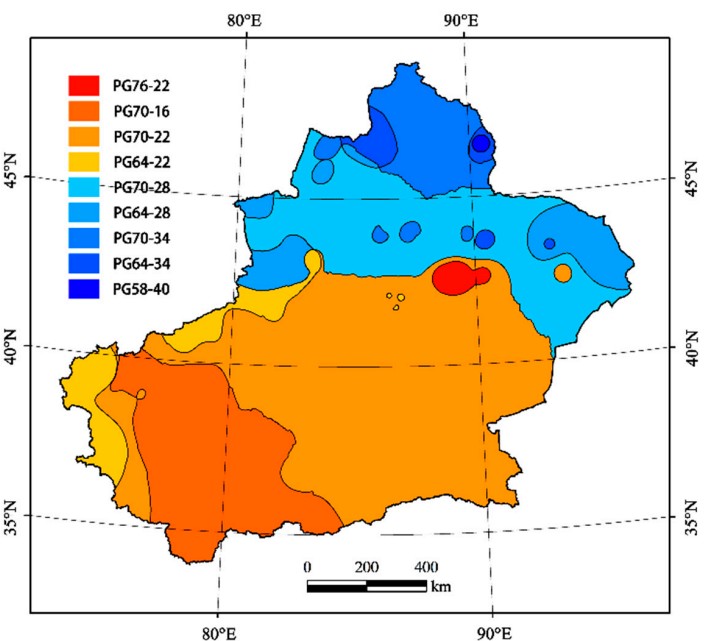

**Figure 4.** Asphalt pavement performance grading map in the Xinjiang region.

## 3. Materials and Methods

### 3.1. Materials

In this paper, four different penetration grades of matrix asphalt binders (90#, 70#, 50#, 30#) and one SBS-modified asphalt binder (modified by 90# matrix asphalt) were researched. All five asphalt binders were provided by PetroChina Karamay Petrochemical Co. SBS (Karamay, China) modified asphalt binder was prepared by a new chemical method. The chemically modified stabilizer is added to make the chemical condensation and crosslinking reaction between SBS-modified asphalt binder and matrix asphalt binder; this improves the compatibility between SBS polymer and matrix asphalt and improves the high-temperature stability and storage stability of SBS-modified asphalt binder. The basic performance indexes of the five asphalt binders are shown in Table 2. The test procedures are operated in accordance with the specification JTG E20-2011 [46], and all the indicators of the five asphalt binders meet the specification JTG F40-2004 [47]. The aggregate used for the asphalt mixture is limestone purchased from the S21 highway project in Xinjiang Province, and its physical indexes are all in line with the application requirements. All parameters of raw materials were tested three times in parallel.

**Table 2.** Basic performance index of five asphalts binders.

| Technical Indexes | Unit | 90# | 70# | 50# | 30# | SBS (I-C) | Limit Values | Standards |
|---|---|---|---|---|---|---|---|---|
| Penetration (25 °C, 100 g, 5 s) | 0.1 mm | 87 (80~100) | 72 (60~80) | 55.8 (40~60) | 32.8 (20~40) | 76 | - | T 0604 |
| Penetration index, PI | - | −1.11 | −0.34 | −0.22 | −0.02 | 0.53 ($\geq$−0.4) | −1.5~+1.0 | T 0604 |
| Softening point, TR&B | °C | 45.3 ($\geq$45) | 49.0 ($\geq$46) | 50.2 ($\geq$49) | 55.6 ($\geq$55) | 65.6 ($\geq$55) | - | T 0606 |
| Ductility (15 °C, 5 cm/min) | cm | >100 ($\geq$100) | >100 ($\geq$100) | >100 ($\geq$80) | 61 ($\geq$50) | / | - | T 0604 |
| Ductility (10 °C, 5 cm/min) | cm | >100 ($\geq$20) | >100 ($\geq$20) | >100 ($\geq$15) | 8 ($\geq$10) | 42.1 ($\geq$30) [a] | - | T 0604 |
| Density@15 °C | g/cm$^3$ | 0.983 | 0.986 | 0.985 | 0.986 | 0.985 | actual measurement | T 0603 |
| Dynamic viscosity@60 °C | Pa·S | 252 ($\geq$160) | 421 ($\geq$180) | 879 ($\geq$200) | 1783 ($\geq$260) | 1.977 ($\leq$3) [b] | - | T 0620 |
| After RTFOT (163 °C 85 min) | | | | | | | | T 0610 |
| Mass change | % | −0.035 | −0.035 | −0.064 | −0.088 | −0.185 ($\leq$1) | $\leq$0.8 | T 0610 |
| Penetration ratio @25 °C | % | 77 ($\geq$57) | 75 ($\geq$61) | 74 ($\geq$63) | 77 ($\geq$65) | 84 ($\geq$60) | - | T 0604 |
| Ductility (15 °C, 5 cm/min) | cm | >100 ($\geq$20) | >100 ($\geq$15) | 27 ($\geq$10) | 9 | / | - | T 0605 |
| Ductility (10 °C, 5 cm/min) | cm | 40 ($\geq$8) | 11 ($\geq$6) | / | / | 25.1 ($\geq$20) [c] | - | T 0605 |

Note: The values in parentheses are the limit values required by the specification. The limit values in Table 2 are the requirements for matrix asphalt. [a] The ductility test of SBS-modified asphalt binder (unaged) provides the results at 5 °C. [b] The viscosity is Brookfield rotational viscosity at 135 °C. [c] The ductility test of SBS-modified asphalt binder (RTFOT-aged) provides the results at 5 °C.

### 3.2. Performance Methods

The performance of five asphalt binders was evaluated according to the Superpave binder specification [48]. Dynamic shear rheometer (DSR) test and bending beam rheometer (BBR) test were used to evaluate the high-temperature rutting resistance, medium-temperature fatigue resistance, and low-temperature cracking resistance of the matrix asphalt and SBS-modified asphalt with different penetration grades. Finally, the PG grading of the asphalt binder was obtained. The PG grading is used to adapt the performance grades of asphalt required for different asphalt pavements in the Xinjiang region.

### 3.2.1. Dynamic Shear Rheometer (DSR)

The high-temperature performance of the five asphalts can be measured by dynamic shear rheometer (DSR). The dynamic shear rheometer (DSR) is shown in Figure 5. The device is manufactured by Ta Co., Ltd. from Austin, TX, USA, model CV0150 AR1500ex. The test parameters, such as the composite shear modulus (G*) and the phase angle (δ), can be obtained by testing the relationship between the applied stress and the measured strain. The PG high-temperature performance of asphalt binder is generally characterized by the rutting factor (G*/sinδ) of the unaged asphalt binder and RTFOT short-term-aged asphalt binder.

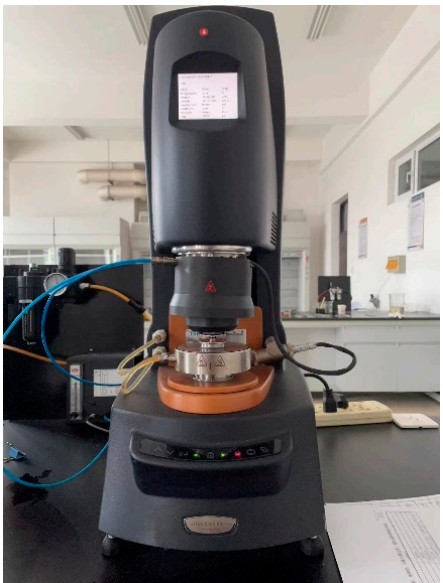

**Figure 5.** Dynamic shear rheometer (DSR).

The test was conducted in the temperature range of 52~88 °C with an increment of 6 °C, and a 25 mm diameter rotor was selected for the parallel plate geometry and sample height equal to 1 mm. When the temperature reaches equilibrium, the equipment is automatically tested at a frequency of 10 rad/s and selected stress target values. The test is automatically terminated when the rutting factor (G*/sinδ) of the unaged asphalt binder is less than 1 kPa and the rutting factor (G*/sinδ) of the RTFOT-aged asphalt binder is less than 2.2 kPa. Each sample was tested once because of the high reproducibility of DSR on rheological properties of asphalt binder.

The intermediate temperature fatigue performance of asphalt is generally characterized by fatigue factor (G*·sinδ) of RTFOT + PAV-aged asphalt binder. PAV aging container is shown in Figure 6. The test temperatures are 10 °C, 16 °C, 22 °C, and 28 °C, and an 8 mm diameter rotor was selected for the parallel plate geometry and sample height equal to 2 mm. Asphalt binder with PG grading of PGm-n requires a fatigue factor (G*·sinδ) of less than 5000 kPa at (m−n)/2 + 4 °C to meet the specification.

### 3.2.2. Bending BEAM Rheometer (BBR)

The bending beam rheology (BBR) test can accurately evaluate the ability of asphalt to resist cracking under low-temperature conditions. The bending beam rheology (BBR) is shown in Figure 7. The device is manufactured by CANNON Co., Ltd. from Melville, NY, USA, model PE-BBR-F. The parameters such as creep stiffness modulus S and m-value of asphalt binder are calculated by applying a constant load and measuring the deflection. The S represents the stiffness of the material, and m-value represents the stress dissipation/relaxation capacity due to temperature change. The lower the S value and the higher the m-value, the better the low-temperature performance of the asphalt binder.

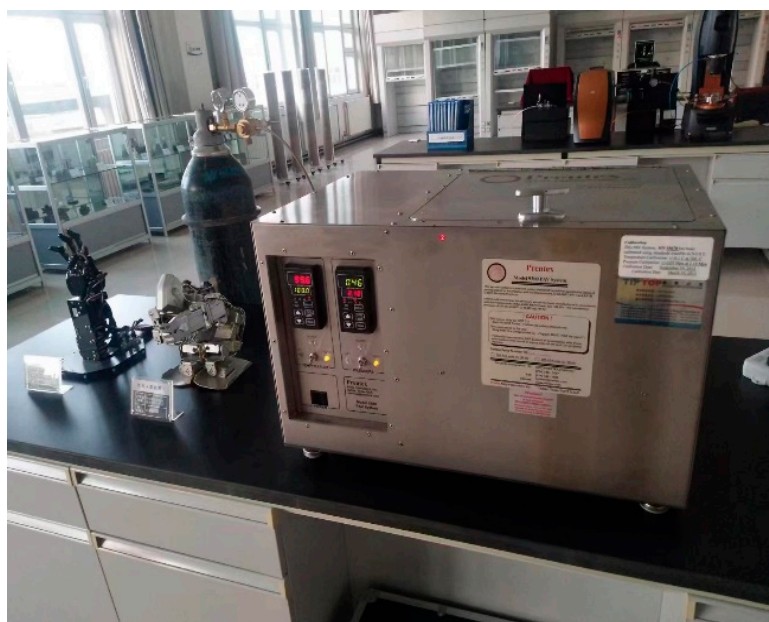

**Figure 6.** PAV aging container.

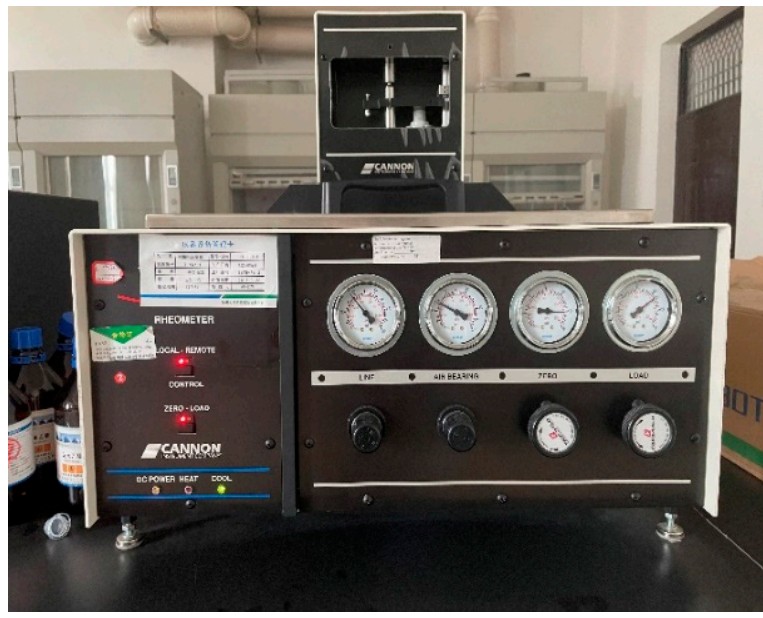

**Figure 7.** Bending beam rheology (BBR).

Trabecular bending samples of five kinds of asphalt binders were prepared, and the sample size was 127 mm × 12.7 mm × 6.35 mm. In the Superpave specification, the test temperature will be increased by 10 °C, and the stiffness modulus with loading time of 60 s will be replaced by the stiffness modulus after loading at the minimum pavement design temperature of 2 h. The specification requires the stiffness modulus (S) with loading time of 60 s to be less than 300 MPa, and m-value is greater than 0.3. At this time, the temperature will be reduced by 10 °C, which is the minimum temperature for asphalt to meet the low-temperature requirements. The tests were conducted at low temperatures of −12 °C, −18 °C, and −24 °C, and the samples were loaded continuously for 240 s. During the test, the creep stiffness modulus (S) and creep rate m-value of trabecular specimens were automatically collected by the computer through the sensor. Two replicate specimens of BBR were tested for each type of asphalt binder studied. For each type of asphalt binder tested, the coefficient of variation was below 15%.

### 3.2.3. Asphalt Mixture Test

The AC-16 asphalt mixtures were prepared regarding the Marshall design method in this work. The gradation composition of AC-16 is listed in Figure 8. According to the preliminary experiments, the ratio of five kinds of asphalt binders to aggregate was set at 4.5% (90#), 4.6% (70#), 4.6% (50#), 4.7% (30#), and 4.7% (SBS).

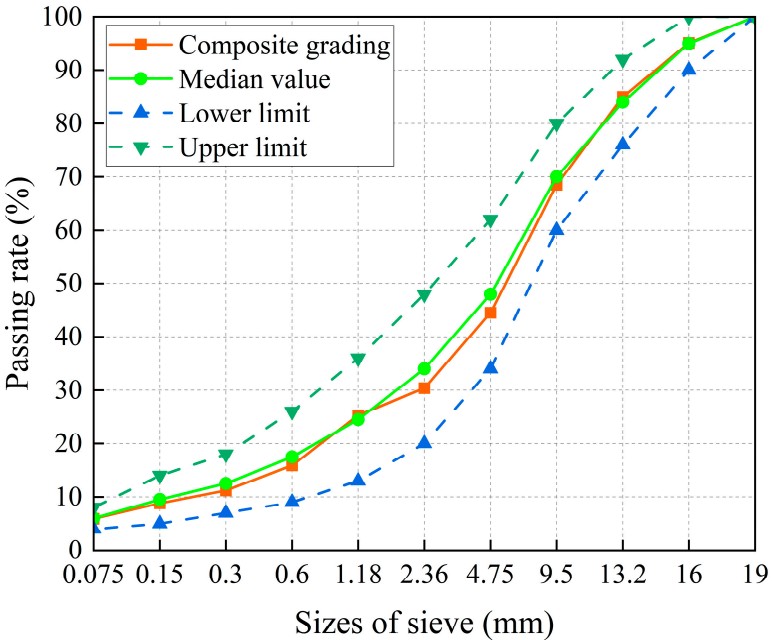

**Figure 8.** The gradation composition of AC-16.

## 4. Results and Discussion

### 4.1. High-Temperature PG

The composite shear modulus (G*) and phase angle (δ) of the five asphalt binders after unaged and RTFOT short-term aging are shown in Figure 9a–d.

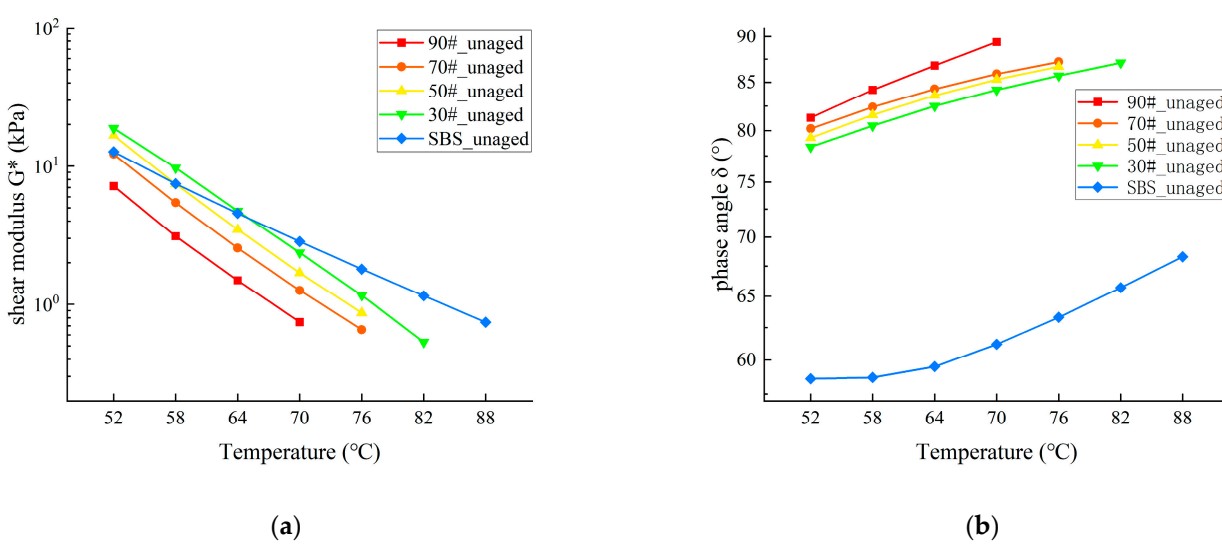

(**a**)                                                                                          (**b**)

**Figure 9.** *Cont.*

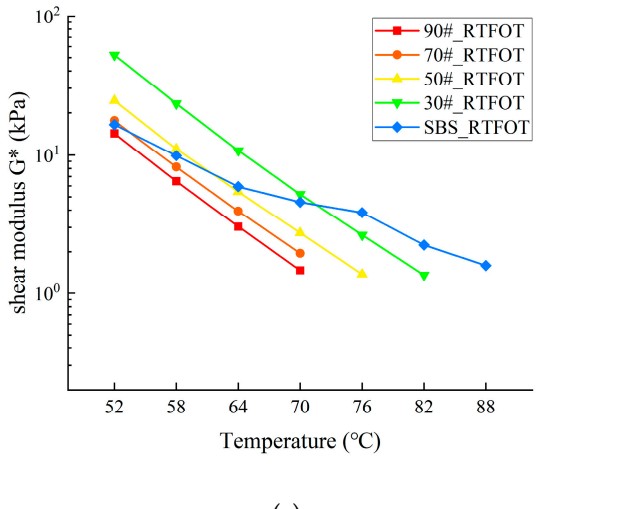
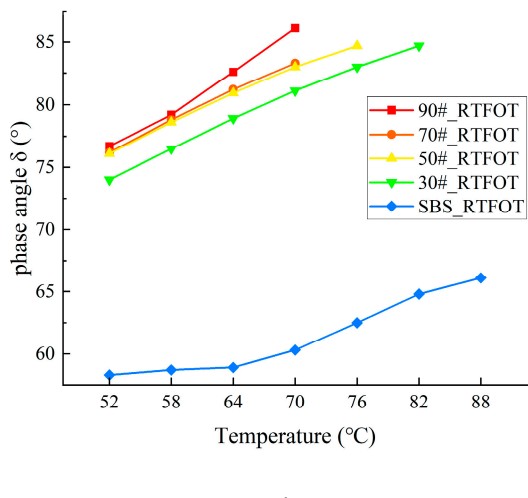

(**c**)    (**d**)

**Figure 9.** The composite shear modulus (G*) and phase angle (δ) at unaged and RTFOT-aged conditions. (**a**) G* at unaged condition; (**b**) δ at unaged condition; (**c**) G* at RTFOT-aged condition; (**d**) δ at RTFOT-aged condition.

The composite shear modulus (G*) represents a measure of the total resistance of the material during repeated shear deformation. The larger the value of G*, the stronger the ability of the asphalt to resist shear deformation. The lower the penetration grade of the base asphalt, the greater the G* value, and the unaged SBS-modified asphalt has better shear deformation resistance than the matrix asphalt binder when the temperature reaches 70 °C. Phase angle (δ) indicates the proportion of viscous components of asphalt binder; the larger the phase angle (δ), the larger the proportion of viscous components of asphalt binder. The smaller the proportion of elastic components, the weaker the ability to recover after deformation. The large difference in phase angle δ between SBS-modified asphalt binder and matrix asphalt binder is mainly due to the addition of the SBS modifier, which makes the asphalt binder exhibit more elastic properties.

After experiencing RTFOT short-term aging, the composite shear modulus G* of all five asphalt binders increased to varying degrees, and 30# is the most obvious, indicating that short-term aging increases the ability of the asphalt binder to resist shear deformation. At the same time, the phase angle decreases to varying degrees, indicating that short-term aging will increase the proportion of elastic components in the asphalt binder. The phase angle of SBS-modified asphalt has the smallest variation, which indicates that the addition of SBS-modified asphalt improves the anti-aging property of asphalt. The matrix asphalt of Karamay has excellent properties, but it is difficult to modify. The SBS-modified asphalt used in this paper is prepared by a new chemical method. Although some stabilizers and regulators are added to improve the stability to a certain extent, it is generally not as stable as the matrix asphalt. Therefore, after RTFOT aging, the performance of the SBS-modified asphalt appears to have certain deviations at different temperatures. In summary, short-term aging will enhance the ability of the asphalt binder to resist rutting deformation and has the greatest effect on the asphalt binder of a low penetration grade.

The rutting factor (G*/sinδ) is used in the Superpave specification to determine the high PG temperature of the asphalt binder. The benchmarks for rutting parameters equal to or higher than 1.0 kPa and 2.2 kPa are adopted for unaged and RTFOT-aged, respectively. The rutting factors of the five asphalt binders are shown in Figure 10a,b.

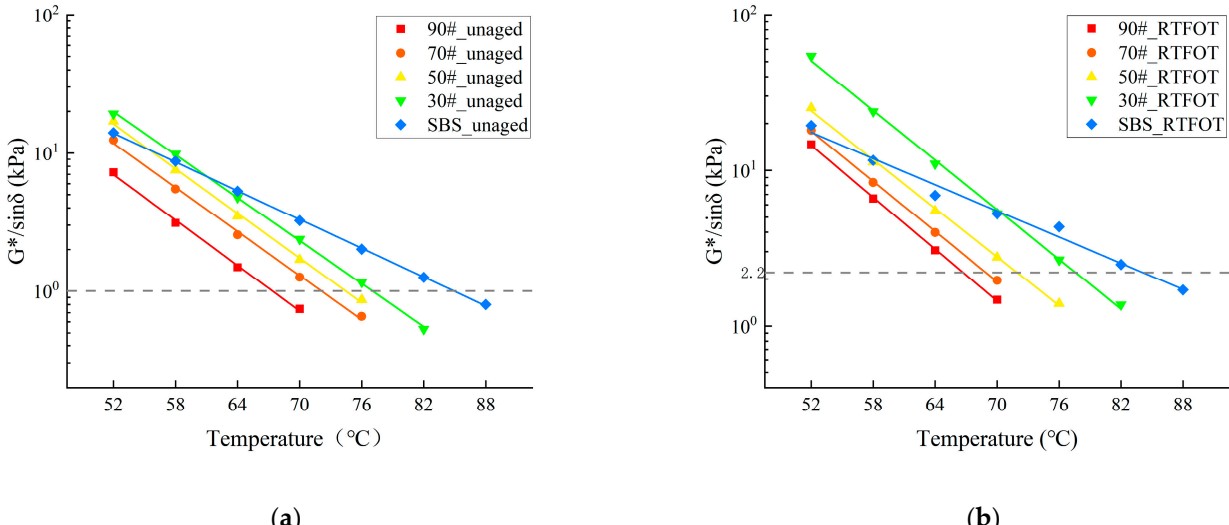

(**a**)                  (**b**)

**Figure 10.** The rutting factor ($G^*/\sin\delta$) of five asphalt binders. (**a**) Unaged; (**b**) RTFOT-aged.

As shown in Figure 10a,b, the high-temperature PG of the five asphalt binders (90#, 70#, 50#, 30#, and SBS) are 66.7 °C, 69.0 °C, 72.1 °C, 76.8 °C, and 83.2 °C, respectively. The high-temperature rutting resistance of the five asphalt binders is SBS > 30# > 50# > 70# > 90#.

Compared with 90# matrix asphalt, the continuous gradation temperature of 70#, 50#, and 30# increased by 2.3 °C, 5.4 °C, and 10.1 °C, respectively, mainly because there are more elastic components in the asphalt binder with low penetration grade, while the continuous gradation temperature of SBS-modified asphalt increased by 16.5 °C. It shows that the SBS modifier can significantly improve the high-temperature rutting resistance of the asphalt binder.

### 4.2. Intermediate Temperature PG

The composite shear modulus ($G^*$) and phase angle ($\delta$) of the five asphalt binders under RTFOT + PAV aging conditions are shown in Figure 11a,b.

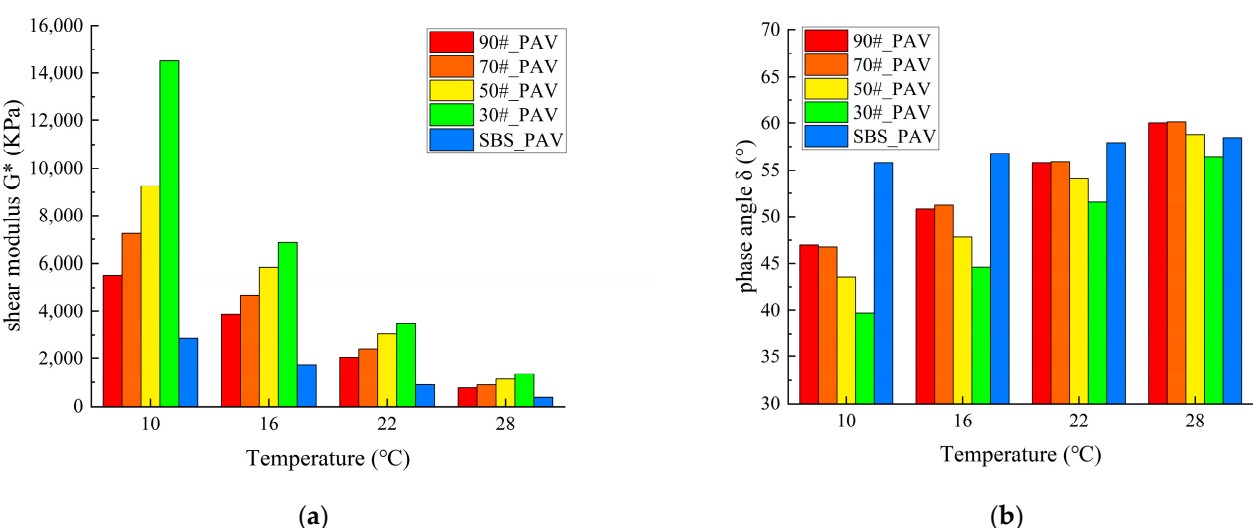

(**a**)                  (**b**)

**Figure 11.** The rutting factor ($G^*/\sin\delta$) of five asphalt binders. (**a**) Unaged; (**b**) RTFOT-aged.

From Figure 11a,b, it can be seen that the composite shear modulus of the five asphalt binders gradually decreases with increasing temperature, and the phase angle gradually becomes larger with increasing temperature. The lower the penetration grade of the matrix

asphalt, the higher the composite shear modulus and the smaller the phase angle, which is consistent with the law of unaged asphalt under high-temperature conditions.

Compared with the matrix asphalt, the SBS-modified asphalt binder has a lower composite shear modulus, and the phase angle is greater than the remaining four matrix asphalt binders before reaching 25 °C. After the temperature reaches 25 °C, the phase angle is gradually smaller than that of each matrix asphalt binder, and the phase angle of SBS-modified bitumen changes the least at the four temperatures, indicating that its ability to resist shear deformation is less affected by temperature.

According to the Superpave specification, the fatigue factor ($G^*\cdot\sin\delta$) was used as the fatigue parameter of the asphalt binder, as shown in Figure 12, which was limited to 5000 kPa as the performance criterion of RTFOT + PAV aging binder at intermediate temperatures. It can be observed that at the intermediate temperature, the fatigue factor size of the five asphalt binders is regular, SBS > 90# > 70# > 50# > 30#, indicating that the lower the penetration grade of the matrix asphalt, the less flexible the asphalt is, which also shows that SBS-modified asphalt has better fatigue performance than the matrix asphalt.

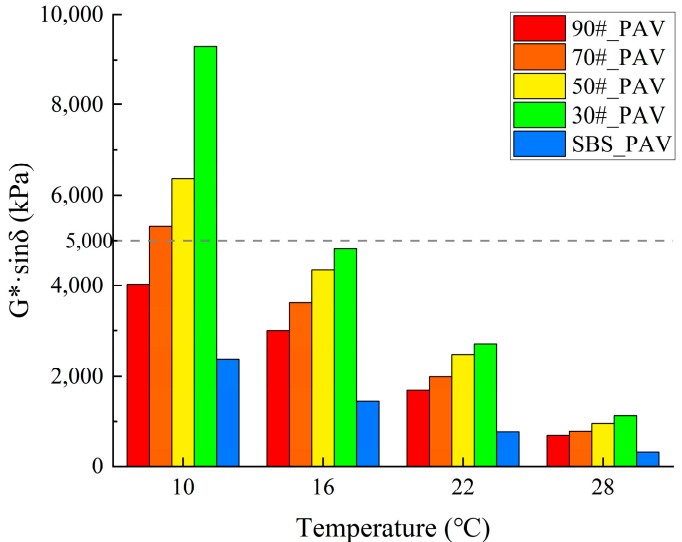

**Figure 12.** The fatigue factor ($G^*\cdot\sin\delta$) of five asphalt binders.

Overall, the fatigue factors of all five asphalt binders are much less than 5000 kPa at the regulated temperature, indicating that the asphalt binder from Karamay has good fatigue resistance.

*4.3. Low-Temperature PG*

The creep stiffness modulus S and m-values calculated from the BBR tests at three different temperatures (−12 °C, −18 °C, and −24 °C) with a loading time of 60 s are shown in Figure 13a,b, respectively. As shown in Figure 13a,b, the 90# and 70# asphalt binders meet the Superpave requirement at −18 °C, 50# and 30# asphalt binders meet the Superpave requirement at −12 °C, and SBS-modified asphalt reaches the Superpave limit at the temperature of −24 °C.

The lower the modulus of stiffness, the lower the temperature stress of the material under the same temperature shrinkage strain, indicating that the low-temperature cracking resistance of the material is stronger. At the same temperature, among the five asphalts, SBS-modified asphalt has the lowest stiffness modulus, and 30# asphalt has the highest stiffness modulus. The low-temperature cracking resistance of the five asphalt binders is SBS > 90# > 70# > 50# > 30#. Compared with 90# base asphalt, the continuous grading temperature of 70#, 50#, and 30# base asphalt decreased by 1.8 °C, 4.6 °C and 7.6 °C, respectively, mainly because of the increase of elastic components in the low penetration grade asphalt binder, while the continuous grading temperature of SBS-modified asphalt

increased by 4.3 °C, which indicates that the addition of SBS modifier can improve the low-temperature performance of asphalt binder.

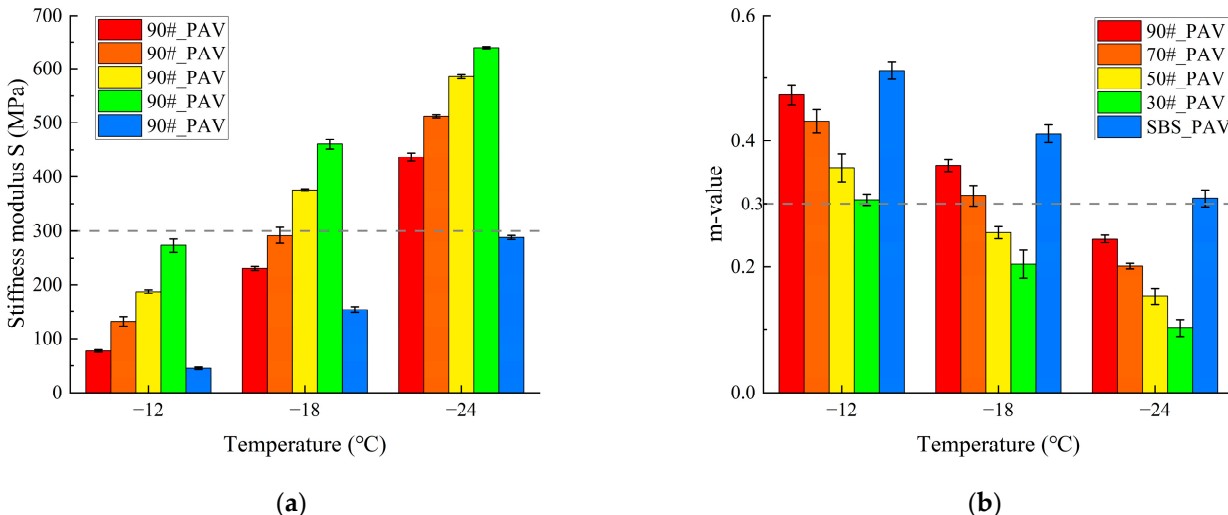

(**a**)  (**b**)

**Figure 13.** The creep stiffness modulus S and m-values of five asphalt binders. (**a**) Stiffness modulus S; (**b**) m-values.

### 4.4. Intermediate Temperature PG

The full performance grades, continuous performance grades, and the difference between continuous high PG and continuous low PG of the five asphalt binders evaluated in this paper are shown in Table 3.

**Table 3.** PG grading of five asphalt binders.

| Parameter | Condation | Specfication | PG | | | | |
|---|---|---|---|---|---|---|---|
| | | | **90#** | **70#** | **50#** | **30#** | **SBS** |
| G*/sinδ | Original | ≥1 kPa | 67.5 | 71.8 | 74.4 | 76.8 | 84.9 |
| G*/sinδ | RTFOT | ≥2.2 kPa | 66.7 | 69.0 | 72.1 | 77.6 | 83.2 |
| | Upper PG | | 64 | 64 | 70 | 76 | 82 |
| G*·sinδ | PAV | ≤5000 kPa | 22 | 22 | 28 | 31 | 28 |
| | Intermediate PG | | 22 | 22 | 28 | 31 | 28 |
| S | PAV | ≤300 MPa at 60 s | −30.0 | −28.2 | −25.6 | −22.9 | −34.3 |
| m-value | PAV | ≥0.3 kPa at 60 s | −31.3 | −28.7 | −25.4 | −22.4 | −34.5 |
| | Lower PG | | −28 | −28 | −22 | −22 | −34 |
| | PG | | 64–28 | 64–28 | 70–22 | 76–22 | 82–34 |
| | Continuous PG | | 66.7–30 | 69–28.2 | 72.1–25.4 | 76.8–22.4 | 83.2–34.3 |
| Difference between cont. high and cont. low PG | | | 96.7 | 97.2 | 97.5 | 99.2 | 117.5 |

For the matrix asphalt binder, the lower the penetration grade, the better the high-temperature performance and the higher the PG high-temperature grade, but the difference between high continuous PG and low continuous PG for different matrix asphalt is not significant, indicating that the matrix asphalt has improved its high-temperature performance while the low-temperature performance also decreases accordingly.

According to the full performance grade for classification, 90# and 70# asphalt binders in the same PG range, indicating that in some cases, the PG grading for asphalt performance

is not fine enough to distinguish; it is recommended to combine the penetration grade and continuous PG classification together for reference.

Due to the addition of a modifier, SBS-modified asphalt can not only improve the high-temperature performance but also improve the low-temperature performance; thus, it has a larger continuous PG span and can adapt to a wider temperature domain. In the area with a larger temperature difference, SBS-modified asphalt can be considered.

### 4.5. Durability of Asphalt Mixtures

The properties of the five asphalt mixtures are shown in Table 4. The durability of asphalt pavement can be characterized by high-temperature performance, low-temperature performance, moisture susceptibility, and fatigue performance. By analyzing Table 4, we find that an asphalt mixture with low penetration has better rutting resistance and water stability, but low-temperature crack resistance and fatigue performance are poor, and it can only meet the requirements of the code in some areas. The dynamic stability, ultimate tensile strain, residual Marshall stability, and fatigue life of the SBS-modified asphalt mixture are 4.7 times, 0.93 times, 1.11 times, and 2.10 times those of the 90# matrix asphalt binders, respectively. To sum this up, the SBS-modified asphalt mixture has better durability.

**Table 4.** Performance of five asphalt mixtures.

| Items | High-Temperature Performance | Low-Temperature Performance | Moisture Susceptibility | | | Fatigue Performance |
|---|---|---|---|---|---|---|
| | Dynamic Stability (Times/mm)/Std.Dev | Ultimate Flexural Strain/Std.Dev | $MS_1$ (kN) /Std.Dev | MS (kN) /Std.Dev | RMS (%) | Fatigue Life (Times)/Std.Dev |
| 90# | 1435/127.72 | 2818/108.22 | 11.46/0.76 | 14.12/0.63 | 81.2 | 61,220/3869 |
| 70# | 2039/130.50 | 2611/84.25 | 14.70/0.58 | 17.89/0.72 | 82.3 | 49,524/3325 |
| 50# | 2703/75.68 | 2383/78.55 | 16.55/0.74 | 19.07/0.84 | 86.8 | 34,667/1845 |
| 30# | 3514/77.31 | 2151/69.17 | 18.17/0.92 | 20.79/1.06 | 87.4 | 30,315/1926 |
| SBS | 8183/310.95 | 2628/83.42 | 16.73/0.75 | 18.64/0.88 | 89.8 | 128,769/8871 |

## 5. Summary and Conclusions

By drawing a performance grading map of asphalt pavement in Xinjiang and researching the PG grading of five asphalt binders in Karamay, with the aim to provide guidance and reference for the selection of asphalt in different areas of Xinjiang to ensure that it is more adaptable to local climatic conditions, the following conclusions were obtained according to the results of the study:

(1) The asphalt pavement performance grading map of Xinjiang region divides Xinjiang into nine sub-districts, which indicates that the climate varies significantly in different areas of Xinjiang. The four partitions with the largest area share are PG70-16, PG70-22, PG70-28, and PG70-34, indicating that the pavement temperature is close to 70 °C in most areas of Xinjiang during the high-temperature season.

(2) For the five partitions with a continuous PG range over 92 °C (PG76-22, PG70-28, PG70-34, PG64-34, and PG58-40), modified bitumen is recommended to ensure that the pavement performance needs can be met. The remaining four subdivisions are recommended to use matrix asphalt to meet the performance requirements in order to achieve economic and environmental protection.

(3) The lower the needle penetration grade of the matrix asphalt, the better the high-temperature performance, and the worse the low-temperature performance, but overall the continuous PG span difference is not large; SBS-modified asphalt continuous PG span can be higher than the matrix asphalt by about 20 °C.

(4) In the case that different penetration grades of asphalt have the same PG grading, it is recommended to combine the penetration grade and continuous PG range together for reference.

(5) By combining the PG grades of five kinds of asphalt with the performance zoning map of the Xinjiang region, we find that 70# asphalt can adapt to most areas of northern

Xinjiang, and 50# asphalt can adapt to most areas of southern Xinjiang. Additionally, 90# and 30# are only useful in some areas. SBS-modified asphalt can not meet the requirements only in the PG58-40 area.

(6) An asphalt mixture with a low penetration has better rutting resistance and water stability, but low-temperature crack resistance and fatigue performance are poor; only in some areas can it meet the requirements of the code, and an SBS-modified asphalt mixture has better durability. According to the road temperature, the paper examined the PG grading of different asphalt binders, which provides useful information for the selection of bitumen in different areas of Xinjiang. However, due to the lack of pavement temperature data, the pavement temperature used in this study is converted by air temperature, so the monitoring and collection of pavement temperature data will be very important work in the future. Additionally, not many types of asphalt were studied in this paper. In order to provide more options for the applicability of asphalt binder in different areas of Xinjiang, PG grading studies on different types of asphalt binder from more manufacturers are needed in the future.

**Author Contributions:** Conceptualization, L.F.; validation, L.F.; formal analysis, C.D.; resources, L.F.; writing—original draft preparation, C.D.; writing—review and editing, C.D.; supervision, L.F.; project administration, Y.X.; funding acquisition, L.F. and Y.X. All authors have read and agreed to the published version of the manuscript.

**Funding:** This research was funded by "customized" asphalt road performance of Xinjiang Academy of Transportation Research, grant number 2021-KT-01.

**Institutional Review Board Statement:** Not applicable.

**Informed Consent Statement:** Not applicable.

**Data Availability Statement:** Data will be made available on request.

**Acknowledgments:** We would like to thank the anonymous reviewers for their constructive feedback and detailed suggestions.

**Conflicts of Interest:** The authors declare no conflict of interest.

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
