# Peer review of "Performance Zoning of Asphalt Pavement and Performance Grade (PG) of Asphalt Binder in Karamay: A Case Study of Xinjiang, China"

_sustainability, doi:10.3390/su15129742_

Round 1

Reviewer 1 Report

Journal: Sustainability

Manuscript ID: Sustainability 2023, 13, x

Manuscript Type: Article

Manuscript title: Research on Performance Zoning of Asphalt Pavement in Xinjiang and Performance Grade (PG) of Asphalt Binder in Karamay

Considering the above-aforementioned manuscript, the following query should be considered for the manuscript to be comprehensible by the authors.

Query and advise

1-     Future work should be included.

2-     The flowchart must be given to the manuscript

3-     Limit values and standards should be shown in the tables.

4-     What type of device was the dynamic shear rheometer (DSR) test performed with and what is the standard?

5-     The total number of tests and the number of samples administered should be specified.

6-     Which standards were used to determine the properties of the mixtures used in the experimental study?

7-     How was the SBS rate determined?

8-     It should be stated what is The main reason for the SBS deviation in Figure 7.

9-     SBS properties should be given in the manuscript.

10-  Doesn't using the SBS modifier show hardness with the aging effect in the asphalt binder? (page 11, lines 309-310)

11-  The threshold value should be indicated in Figure 9 and Figure 10.

12-  Page 13 lines 349-350 should be reviewed. Doesn't the use of SBS-modified asphalt increase the shear module value?

13-  ArcGIS should be explained in detail.

14-  What is the type of mapping?

15-  What are the criteria for creating isotherm curves?

16-  Is point data not considered in the mapping?

17-  Which years were simulated in long and short-term aging?

18-  Where has modified bitumen been used as an application in real conditions?

The study is an experimental study and the results are satisfactory. The authors should consider the above suggestions and questions to better understand the research work. Therefore, minor corrections are required.

Journal: Sustainability

Manuscript ID: Sustainability 2023, 13, x

Manuscript Type: Article

Manuscript title: Research on Performance Zoning of Asphalt Pavement in Xinjiang and Performance Grade (PG) of Asphalt Binder in Karamay

Considering the above-aforementioned manuscript, the following query should be considered for the manuscript to be comprehensible by the authors.

Query and advise

1-     Future work should be included.

2-     The flowchart must be given to the manuscript

3-     Limit values and standards should be shown in the tables.

4-     What type of device was the dynamic shear rheometer (DSR) test performed with and what is the standard?

5-     The total number of tests and the number of samples administered should be specified.

6-     Which standards were used to determine the properties of the mixtures used in the experimental study?

7-     How was the SBS rate determined?

8-     It should be stated what is The main reason for the SBS deviation in Figure 7.

9-     SBS properties should be given in the manuscript.

10-  Doesn't using the SBS modifier show hardness with the aging effect in the asphalt binder? (page 11, lines 309-310)

11-  The threshold value should be indicated in Figure 9 and Figure 10.

12-  Page 13 lines 349-350 should be reviewed. Doesn't the use of SBS-modified asphalt increase the shear module value?

13-  ArcGIS should be explained in detail.

14-  What is the type of mapping?

15-  What are the criteria for creating isotherm curves?

16-  Is point data not considered in the mapping?

17-  Which years were simulated in long and short-term aging?

18-  Where has modified bitumen been used as an application in real conditions?

The study is an experimental study and the results are satisfactory. The authors should consider the above suggestions and questions to better understand the research work. Therefore, minor corrections are required.

Reviewer 2 Report

In this study, a performance grading map of asphalt pavement in Xinjiang region were established based on 11 years of temperature data and exploration of asphalt mixture properties. It is of great merit to guide the selection of asphalt binder in pavement design in Xinjiang Province. The manuscript can be improved by the following points:

(1) Some format problems need to be fixed:

(a) In the abstract, there is a situation of inconsistent tenses, and it is suggested to check and modify in detail.

(b) Some tables are not on the same page like table 3.

(c) Figures 7 through 11 need to be adjusted.

(2) In the second section, Inverse Distance Weighting (IDW) interpolation method is used to generate performance grading map in the Xinjiang region. We know that there are various methods of spatial interpolation, and why IDW is used in the paper instead of other methods such as Ordinary Kriging or Collaborative Kriging? Please explain the reason for using the IDW interpolation method.

(3) Whether only one set of data is used to obtain the test results in this paper? If only one set of data is used, how can the reliability of the test results be proved? If multiple sets of data are used, how can they be expressed and what is the error?

(4) The analysis of the results of the five asphalt binders test is not comprehensive enough and should be carefully analyzed and discussed in relation to Section 2.

Minor editing of English language is required

Reviewer 3 Report

This is interesting and timely work. Some comments:

1.       This is not a research paper, please change the title to ‘case study’.

2.       Abstract: do the authors propose any new penetration grade ranges?

3.       To use PG system to replace penetration system have been implemented for more than 30 years, and it is working very well. What is new in this study?

4.       Line 117: according to my knowledge, penetration values are not enough to grade PmB, at least softening point temperature are needed. Please revise it.

5.       The testing plan and analysis are reliable.

6.       What is the future perspective, I mean how to correlate the binders’ properties to mixtures?

No

Reviewer 4 Report

The article “Research on Performance Zoning of Asphalt Pavement in Xin- 2 jiang and Performance Grade (PG) of Asphalt Binder in Karamay”, the authors presented durability studies mainly considered temperature as variable. The studies highlighted the drawbacks and to overcome the drawback new pavement method using SBS modifiers performance grade (PG). The following few points must consider to improve the quality of manuscript.

1.The reviews with respect to climatic factors as well as different spatial factors are much required. The authors majorly expressed only few factors with respect to asphalt pavement. The latest review of modified pavement design factors are missing. It is advised to add few latest reviews with respect to climatic, spatial & durability factors with respect to modified pavement deign.

2. Figure-2 & Figure-3 ArcGIS image legend is not clear. It is advised to replace Figure-2 with clear visibility of legend.

3.It is advised to replace all figures. Figures are not clear having visibility issues.

4.The figure-1 technical explanation for low & high temperature is completely missing. It is advised to highlight technical reason in this section.

5.The technical explanation is required for figure-2.

6.The complete durability analysis is based on empirical equations so it is advised to project in abstract section. The slight revision of abstract with respect to this is required.

7.The data analyses with the help of GIS graphs are projected. The complete technical & realistic reasons are missing in each section. It is advised to highlight the data section with technical evidence for durability studies in each section.

8.The asphalt new modified pavement how effective with respect to durability is required to be explained in separate paragraph.

Minor editing of English language required.

Round 2

Reviewer 3 Report

Thanks to the authors' hard work, almost all my comments were revised. No further review process is needed.

No.

Reviewer 4 Report

Suggested Reviews incorporated.

Moderate English corrections required.